# Rapid Optical Biosensing of SARS-CoV-2 Spike Proteins in Artificial Samples

**DOI:** 10.3390/s22103768

**Published:** 2022-05-16

**Authors:** Ying Tao, Sumin Bian, Pengbo Wang, Hongyong Zhang, Wenwen Bi, Peixi Zhu, Mohamad Sawan

**Affiliations:** 1College of Pharmaceutical Sciences, Zhejiang University of Technology, Hangzhou 310014, China; 2111907032@zjut.edu.cn; 2CenBRAIN Lab, School of Engineering, Westlake University, Hangzhou 310024, China; biansumin@westlake.edu.cn (S.B.); wangpengbo@westlake.edu.cn (P.W.); zhanghongyong@westlake.edu.cn (H.Z.); sawan@westlake.edu.cn (M.S.); 3Key Laboratory of Structural Biology of Zhejiang Province, School of Life Science, Westlake University, Hangzhou 310024, China; biwenwen@westlake.edu.cn; 4Institute of Biology, Westlake Institute for Advanced Study, Hangzhou 310024, China

**Keywords:** SARS-CoV-2, fiber-optic biolayer interferometry, biosensor, spike proteins, signal amplification, rapid detection

## Abstract

Tests for SARS-CoV-2 are crucial for the mass surveillance of the incidence of infection. The long waiting time for classic nucleic acid test results highlights the importance of developing alternative rapid biosensing methods. Herein, we propose a fiber-optic biolayer interferometry-based biosensor (FO-BLI) to detect SARS-CoV-2 spike proteins, extracellular domain (ECD), and receptor-binding domain (RBD) in artificial samples in 13 min. The FO-BLI biosensor utilized an antibody pair to capture and detect the spike proteins. The secondary antibody conjugated with horseradish peroxidase (HRP) reacted with the enzyme substrate for signal amplification. Two types of substrates, 3,3′-diaminobenzidine (DAB) and an advanced 3-Amino-9-ethylcarbazole (i.e., AMEC), were applied to evaluate their capabilities in enhancing signals and reaching high sensitivity. After careful comparison, the AMEC-based FO-BLI biosensor showed better assay performance, which detected ECD at a concentration of 32–720 pM and RBD of 12.5–400 pM in artificial saliva and serum, respectively. The limit of detection (LoD) for SARS-CoV-2 ECD and RBD was defined to be 36 pM and 12.5 pM, respectively. Morphology of the metal precipitates generated by the AMEC-HRP reaction in the fiber tips was observed using field emission scanning electron microscopy (SEM). Collectively, the developed FO-BLI biosensor has the potential to rapidly detect SARS-CoV-2 antigens and provide guidance for “sample-collect and result-out on-site” mode.

## 1. Introduction

Recently, widespread Corona Virus Disease 2019 has caused a significant health crisis for the public and the health care system [1]. The latest data from the World Health Organization (WHO) reported over 4.6 hundred million confirmed cases of SARS-CoV-2 globally. The SARS-CoV-2 virus consists of four major structural proteins: spike surface protein, envelope protein, membrane protein, and nucleocapsid protein [2]. In addition, SARS-CoV-2 has a positive-sense single-stranded genomic RNA (Figure 1A). The spike protein, namely the extracellular domain (ECD), facilitates SARS-CoV-2 attacking human cells by using its receptor-binding domain (RBD) to bind with the angiotensin-converting enzyme 2 (ACE2) [3]. RBD, including S1 and S2 subunits [4], mediates receptor binding and membrane fusion reaction [5]. Thus, ECD and RBD are the most significant antigens related to SARS-CoV-2. 

Based on the transducer, the current methods applied to detect SARS-CoV-2 antigens can be divided into two major groups: optical and electrochemical. A field-effect transistor-based biosensor can detect SARS-CoV-2 spike protein at concentrations of 100 fg/mL on a Human Nasopharyngeal Swab [6]. A square wave voltammetry (SWV)-based biosensor can detect SARS-CoV-2 spike protein at 0.11 ng/mL in 30 min [7]. In a recent study, an optical biosensor detected SARS-CoV-2 RBD at the concentration of 12.5 nM in just 5 s with near-infrared technology [8]. Despite the ultra-sensitivity of electrochemical biosensors, optical type biosensors could often decrease the external disturbances to obtain high-intensity optical signals. Meanwhile, they also have great potential in full automation and high throughput, which is crucial in large-scale diagnoses of SARS-CoV-2 in the community [9]. 

To date, various optical biosensors have been developed for quick SARS-CoV-2 detection. The recognition element of an optical biosensor can be classified into aptamer, molecularly imprinted polymer (MIP), and antibody. Tabrizi et al. detected RBD with the use of an MIP as capture, allowing the detection of RBD in the concentration range between 2 and 40 pg/mL with a limit of detection (LoD) of 0.7 pg/mL [10]. Cennamo et al. detected RBD at the concentration of 37 nM by using aptamer as the recognition element [11]. Awada et al. developed an antibody-based biosensor for RBD detection with an LoD of 1 pM [12]. 

In this work, we used a fiber-optic biolayer interferometry (FO-BLI)-based biosensor for on-site and automated detection of SARS-CoV-2 within 13 min. The FO-BLI detection system was described in detail in our two recent papers [13,14]. When vertical light enters, the reflected light from the 1st and 2nd interface of sensors forms an interference curve. The curve moves in the direction of large wavelengths, along with the increase in the thickness of the biolayer of sensors. To achieve SARS-CoV-2 detection, a specific biotinylated neutralizing antibody (MA-RBD) was loaded on the fiber to target ECD and RBD in buffer and artificial samples. The SA biosensors (streptavidin) were commercial fibers that were coated with streptavidin previously. After that, binding between SARS-CoV-2 antigens and the specific secondary anti-S1 antibody conjugated with horseradish peroxidase (MA-S1-HRP) occurred, followed by the interaction between HRP and its substrate to generate sufficient optical signals (Figure 1B). 3,3′-diaminobenzidine (DAB), a common substrate to HRP often used in biology, has proven to be an appropriate enhancer for generating signals in the FO-BLI system (Figure 1C). Considering the potential toxicity of DAB, this work also aimed to search for an alternative enhancer for the FO-BLI technique to achieve both desirable assay performance and environmental and human friendliness. From this perspective, this work utilized an advanced product of 3-amino-9-ethylcarbazole, namely AMEC, to interact with HRP to further improve the optical signals. Similar to DAB, in AMEC staining, AMEC was oxidized by hydrogen peroxide in a reaction typically catalyzed by HRP to form a precipitate at the site of HRP. However, data regarding AMEC-based signal enhancement in the FO-BLI system has rarely been reported. The chemical structure of AMEC is still secretive about the details. In this study, we compared the assay performance for SARS-CoV-2 detection using the two enhancers. The outperformed enhancer was further evaluated regarding its clinical outcome using artificial saliva and serum. 

## 2. Experimental Section

### 2.1. Materials

All solutions were prepared with deionized water purified by a Milli-Q. SARS-CoV-2 RBD, ECD, SARS-CoV RBD, MERS-CoV S protein, SARS-CoV-2 N protein, SARS-CoV-2 RBD (N501Y), and SARS-CoV-2 (2019-nCoV) Spike Neutralizing Antibody (40592-R001, referred to as MA-RBD) were purchased from Sino Biological (Beijing, China). The anti-SARS-CoV-2 (S1)-HRP (ATMA10342Mo, referred to as MA-S309) was purchased from AtaGenix (Wuhan, China). The Octet^®^ K2 2-channel system and streptavidin (SA) sensors were purchased from Sartorius Stedim Biotech GmbH (Gottingen, Germany). The 96-well polystyrene black microplates were obtained from Greiner Bio-One GmbH (Shanghai, China). The 3,3′-Diaminobenzidine (DAB) enhanced liquid substrate system tetrahydrochloride, Tween-20, and Bovine Serum Albumin were purchased from Sigma-Aldrich (Shanghai, China). ZebaTM spin desalting columns (7 K MWCO, 0.5 mL) were purchased from Thermo Scientific (Shanghai, China). The biotinylation kit was obtained from Genemore (Suzhou, China). The ImmPACT^®^ AMEC Red Peroxidase Substrate Kit was purchased from VECTOR Laboratories (Burlingame, CA, USA). The phosphate-buffered saline (PBS, 10 mM, pH 7.4), PBS buffer containing 0.02% (*v*/*v*) Tween-20 and 0.1% BSA (referred to as sample diluent (SD) buffer), and SD buffer containing 274 mM NaCl (referred to as high-salt SD buffer) were freshly prepared in house.

### 2.2. Bioconjugation of Monoclonal Antibody (mAb)

To develop the FO-BLI biosensor for specific antigen detection, a neutralizing antibody against RDB (MA-RBD) was biotinylated according to the manual offered by Genemore, the producer of the biotinylation kit. Briefly, MA-RBD was mixed with biotinylation reagents and shaken for 30 min. Desalting columns were used to centrifuge the mixtures at 1500× *g* for 1 min first to switch the storage solution, followed by another 2 min of centrifugation to remove any unbound biotin.

### 2.3. Comparison of AMEC and DAB for Use as Signal Enhancers in the FO-BLI Biosensors

The biotinylated MA-RBD was prepared at 625 ng/mL in PBS. The MA-S309-HRP was prepared at 1 μg/mL in high-salt SD buffer (PBS pH 7.4, 0.02% Tween, 0.1% BSA, 274 mM NaCl). The plate was agitated at 1000 RPM during the entire experiment. Before the binding measurements, the sensors were pre-hydrated in PBS for at least 10 min. Biotinylated MA-RBD was coated on the surface of the SA sensor for a 90 s loading step. After a 60 s baseline step, sensors were dipped into the wells containing ECD with a series of concentrations (0 pM, 36 pM, 72 pM, 144 pM, 288 pM, 576 pM, 720 pM) for 300 s specific binding. After another 60 s baseline step, the ECD-attached fibers were submerged into the well containing MA-S309-HRP. AMEC and DAB, the two common substrates of horseradish peroxidase enzyme (HRP), were used as signal amplifying reagents. The interaction of AMEC/DAB with HRP generated a red precipitate on the surface of biosensors and enhanced the wavelength shifts.

### 2.4. Establishing an AMEC-Based FO-BLI Biosensor for ECD and RBD Detection in Buffer and Artificial Samples

The first antibody was immobilized on SA biosensors (streptavidin) by coupling the biotin to the antibody. Then, antigen ligands (ECD and RBD) were dissolved in buffer, saliva, and serum and diluted into different multiples by high-salt SD buffer, followed by serial dilutions to obtain gradient concentrations. As proved by Bian et al., high-salt SD buffer effectively reduced non-specific binding [14]. Sensors were dipped into high-salt SD buffer and shaken for 1 min at a speed of 400 rpm for washing. After a wash step, sensors were immersed in ligands solution. Finally, ligands-modified sensors were dipped into the 200-fold diluted second-antibody (MA-S309-HRP) after another wash step. The signal was amplified by signal enhancer AMEC. 

RBD and ECD were similarly detected in three types of matrices: buffer, healthy control serum, and artificial saliva. A high concentration of ECD/RBD stock solution was prepared to make: (i) high-salt SD buffer with RBD spiked from 0 to 400 pM: 0, 12.5, 25, 50, 100, 200, 400 pM, high-salt SD buffer with ECD spiked from 0 to 1152 pM: 0, 36, 72, 144, 288, 576, 1152 pM, (ii) 2× diluted human serum with RBD spiked from 0 to 400 pM, and ECD spiked from 0 to 1152 pM and (iii) 2× diluted saliva with the same range for RBD and ECD.

When applying the FO-BLI into a mimicked matrix, such as saliva and serum, the impact of complex components is essential to be considered. To reduce the matrix effect, the dilution factor was evaluated. Both the serum and saliva samples were diluted 2-fold and 4-fold by high-salt SD buffer at 0 and 200 pM and tested for the matrices effect on the measurement of the fiber.

### 2.5. Establishing an AMEC-Based FO-BLI Biosensor for ECD and RBD Detection in Buffer and Artificial Samples

The specificity was evaluated using four other coronavirus antigens, SARS-CoV RBD, MERS-CoV S-Protein, SARS-CoV-2 N-Protein, and SARS-CoV-2 RBD (N501Y) at a concentration of 720 pM spiked in serum and saliva.

### 2.6. Data Analysis

Specific binding curves for the FO-BLI RBD and ECD biosensors were assessed using “one-site: specific binding” in nonlinear regression of GraphPad Prism 9.02 (GraphPad Software, San Diego, CA, USA). Relative standard deviation (RSD) was used to determine if the standard deviation of the result is small or large when compared to the average. The lower limit of quantification (LLoQ) was defined as the lowest concentration of the standard curve reliably measured with an RSD ≤ 20%.

## 3. Results

### 3.1. Comparison of AMEC and DAB Based FO-BLI Biosensors for SARS-CoV-2 ECD Detection

In the developed FO-BLI biosensor for SARS-CoV-2 detection, the first monoclonal antibody (MA-RBD) targeting spike protein was immobilized on streptavidin (SA) biosensors for 1 min, followed by a wash step for 1 min to reduce any unbound or non-specific binding molecules. Then, ECD or pseudovirus was diluted in the buffer to obtain gradient concentrations as the target. Finally, MA-S309-HRP with a dilution of 500-fold in high-salt SD buffer was incubated with the ECD or pseudovirus-attached fibers for 4 min to enable binding. 

Our latest research in monitoring SARS-CoV-2 antibodies and small molecule drugs proved that the combination of HRP with the metal precipitate DAB was sufficient in enhancing the optical signals [13,14]. Similar to those two works, a dilution factor of 200-fold for DAB was also proved to be appropriate for SARS-CoV-2 ECD detection with a limited background interference and sufficient positive signal when ECD is at a concentration of 1.15 nM. According to the product protocol, the AMEC is suggested to be diluted with the following format: 90 μL reagent-1 and 80 μL reagent-2 into 5 mL substrate buffer. Similarly, to ensure sufficient positive shifts while maintaining negligible negative shifts and avoiding an overload of precipitates on sensor tips, a series of AMEC diluted solutions were tested. As shown in Figure 2A, a further dilution of 6-fold was selected for the use of AMEC as an enhancer, indicating a final solution containing 18 μL reagent-1 and 16 μL reagent-2 to 5 mL substrate buffer. When retaining the signals of the blank control (ECD 0 pM) at 2 nm, signals of highest concentration were 11.5 nm using DAB and 15 nm using AMEC. In summary, 6-fold diluted AMEC led to higher positive shifts but lower background shifts with a signal-to-noise ratio of 14.6, as compared to the signal-to-noise ratio of 7.4 obtained from 200-fold DAB-based signal enhancement. 

The slightly better performance in AMEC-based signal enhancement was further reflected by the standard binding curves obtained with ECD concentration ranging from 0–720 pM in buffer (Figure 2B,C). Moreover, the two enhancers resulted in similar LoD when applying the FO-BLI biosensor in SARS-CoV-2 spike pseudovirus. Particularly, the DAB-based biosensor resulted in an LoD of 2 × 10^7^ copies/mL, while the AMEC-based biosensor resulted in an LoD of 6 × 10^7^ copies/mL (Figure 2D). The optimization for the determination of the pseudovirus needs to be optimized in further studies. 

A comparison of AMEC versus DAB-based FO-BLI biosensors for SARS-CoV-2 ECD detection can be found in Table 1, with details. Following the careful comparison, AMEC is selected as a better alternative to replace DAB to enhance the optical signals in following studies for three reasons. First, the AMEC-based FO-BLI biosensor has a better assay performance with a higher signal-to-noise ratio, although it did not contribute to high sensitivity. Second, the metal precipitate AMEC, which contains one more methyl group compared to AEC, is supposed to be less dangerous for operators, being more environmentally and human friendly. Third, AMEC-based precipitates are insoluble in water but soluble in alcohols and dimethylformamide (DMF), which provides us a chance for fiber regeneration in the near future.

### 3.2. Pre-Clinical Validation of the AMEC Based FO-BLI Biosensors for Rapid SARS-CoV-2 Spike Protein

The specificity was evaluated using four proteins of coronavirus SARS-CoV-2 ECD, SARS-CoV RBD, MERS-CoV S-protein, SARS-CoV-2 N-protein, and SARS-CoV-2 RBD (N501Y) at the concentration of 720 pM. The FO-BLI biosensor showed no cross-reactivity with SARS-CoV RBD, MERS-CoV S-protein, SARS-CoV-2 N-protein, and slightly reacted towards SARS-CoV-2 RBD (N501Y), as shown in Figure 3A. 

When applying the FO-BLI biosensor to the serum and the saliva, the matrix infection should be taken into consideration, as the components in serum and saliva impact the sensor performance. Our latest research in monitoring carbamazepine proved that high-salt SD buffer helps reduce the non-specific binding [14]. The dilution factor of the matrix was evaluated at 2-fold and 4-fold. The 2-fold and 4-fold dilution using high-salt SD buffer resulted in a similar performance for the two quality control samples at RBD 0 and 200 pM in serum and saliva compared to pure high-salt SD buffer (Figure 3B).

### 3.3. AMEC-Based FO-BLI Biosensor for Detecting SARS-CoV-2 RBD and ECD in Spiked Saliva and Saliva

Under the optimized dilution factor of mAb -S-HRP, the ultimate shift was around 12 nm (Figure 3). To mimic the detection matrix for SARS-CoV-2 spike proteins, ECD and RBD were spiked into artificial saliva and healthy control serum for detection. Particularly, ECD concentrations ranging from 0 to 1152 pM were detected in saliva and serum, both of which fitted in a nonlinear dose-response model (Figure 4A; R^2^ = 0.996, *n* = 3 in saliva; R^2^ = 0.996, *n* = 3 in serum). The LoD was determined to be 36 pM. Similarly, RBD concentrations ranging from 0 to 400 pM were detected in saliva and serum, both of which fitted well in a nonlinear dose-response model (Figure 4B; R^2^ = 0.986, *n* = 3 in saliva; R^2^ = 0.956, *n* = 3 in serum). The LoD was determined to be 12.5 pM.

Relative standard deviation (RSD) shows the performance of the method for both RBD and ECD detection (*n* = 3). The RSD for the RBD (12.5 pM) was 12.4%, and the RSD for ECD (36 pM) was 14.2%. The RSD values of ECD and RBD in buffer were less than 20%, proving that the lower limit of quantification (LLoQ) was reliable. 

A careful comparison between the various SARS-CoV-2 detection platforms reported in the literature and the platform developed in the study regarding the assay turnaround time, sensitivity, recognition element, transducer, and substrate was performed, as shown in Table 2. The sensor capabilities of our platform include its high speed of detection (sample to result ≤13 min), high throughput (up to 96 samples with the use of an 8-channel system), and higher analytical sensitivity (low LoD), and a flexible protocol of detecting mutant proteins.

### 3.4. Characterization of AMEC Generated Precipitate for Signal Enhancement in FO-BLI Biosensors

On the top of the FO-BLI sensor, the ImmPACT RED, which was combined with H_2_O_2_ and HRP, was oxidized to reddish-brown precipitate (Figure 5). The ImmPACT RED is about 5–10 times greater in sensitivity over 3-amino-9-ethylcarbazole. The crystalline pattern was investigated by field emission scanning electron microscopy (Zeiss Gemini500 SEM) from Zeiss (Jena, Germany). The images showed that the sensor loaded with the first antibody exhibited small snowflake shapes, whereas the sensor with precipitates (due to the reaction between HRP and AMEC) exhibited larger dendritic precipitates.

## 4. Discussion

In this study, the FO-BLI biosensor was developed to detect SARS-CoV-2 spike proteins in artificial serum and saliva. The sandwich-based FO-BLI showed the capacity to detect ECD within the range of 36 to 1152 pM and detect RBD of 12.5 to 400 pM without interference from other coronavirus proteins. The matrix effects from serum and saliva were eliminated by using high-salt SD buffer as the buffer solution. Due to the limitation of the signals produced by proteins’ small molecular sizes, this assay amplified the signals by utilizing the reaction between AMEC and HRP. The reaction generated red precipitates on the surface of fiber tips to enhance the wavelength shifts. The enzymatic biosensors successfully achieved the detection limit of RBD and ECD as low as 12.5 pM and 36 pM, respectively. The detection time was approximately shortened to 13 min without extra modification steps under a 2-fold dilution into a high-salt SD buffer. Considering the protocol’s flexibility, the FO-BLI biosensor can be easily adjusted to detect SARS-CoV-2 variants, including delta and omicron. 

Considering the automated character of the FO-BLI system, the spike protein biosensor has advantages in many sectors, such as speed, automation, and high throughput, making it feasible for on-site SARS-CoV-2 detection. However, studies are needed to improve the sensitivity of the biosensors and validate the clinical utility for SARS-CoV-2 detection using human samples.

## Figures and Tables

**Figure 1 sensors-22-03768-f001:**
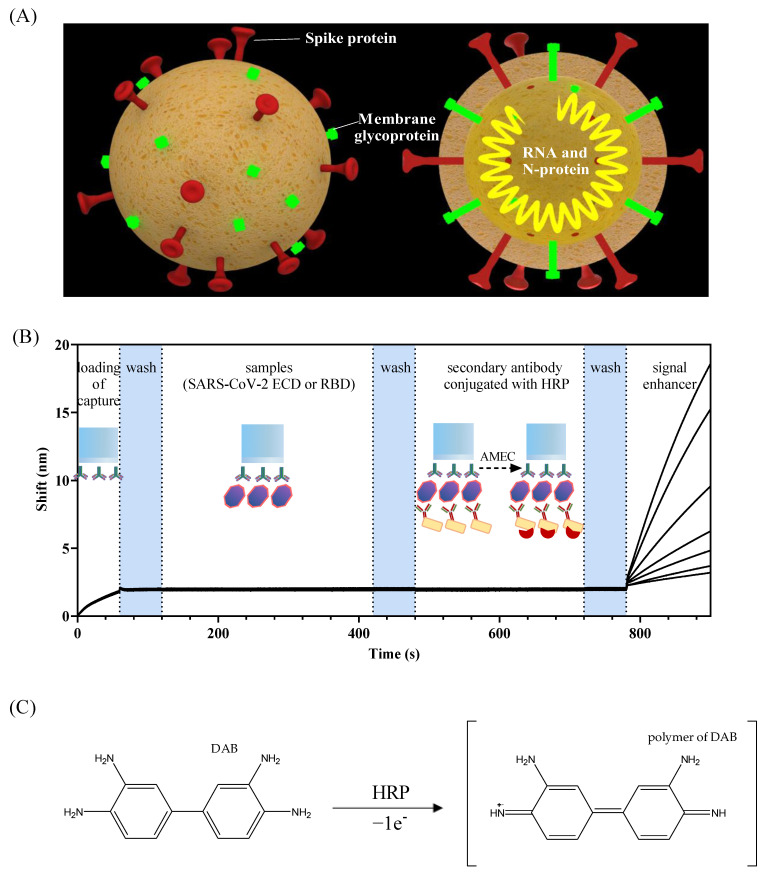
(**A**) The structure of SARS-CoV-2. The FO-BLI biosensor principle and detection flowchart. (**B**) Schematic illustration of the detection flowchart, the sandwich-based assay principle, and the oxidation reaction for signal enhancement on top of a fiber tip. (**C**) Scheme of the reaction between DAB with the HRP enzyme.

**Figure 2 sensors-22-03768-f002:**
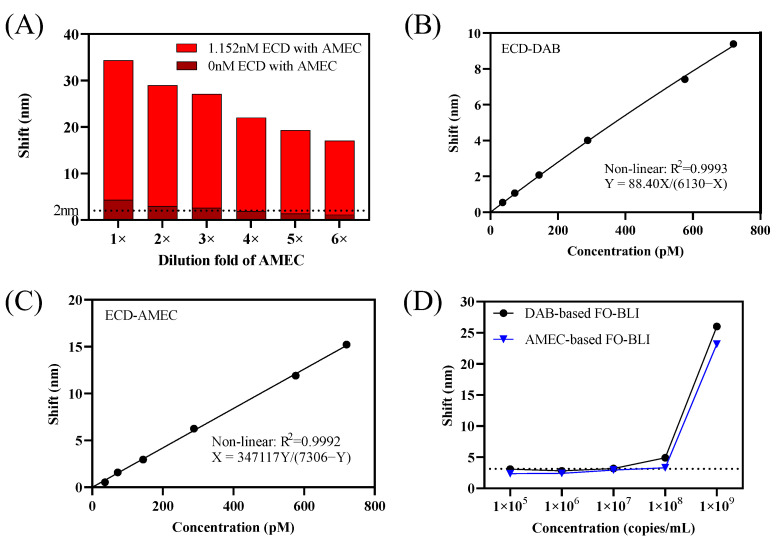
(**A**) Evaluation of the dilution factor of the precipitating agent AMEC on ECD detecting. (**B**,**C**) Nonlinear binding curve of ECD detection (36–720 pM) in high-salt SD buffer using DAB and AMEC, respectively. (**D**) Measurements of six concentrations of pseudovirus based on DAB-based FO-BLI and AMEC-based FO-BLI, respectively.

**Figure 3 sensors-22-03768-f003:**
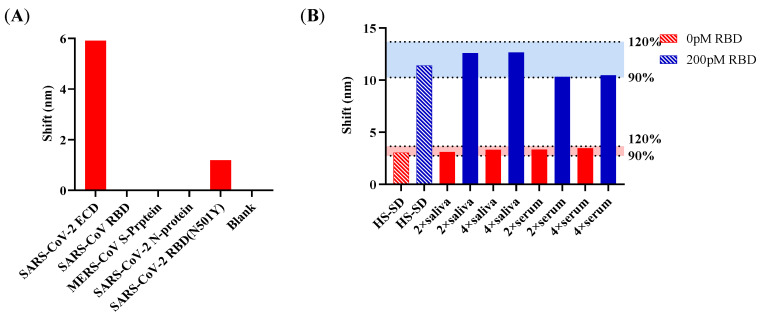
Evaluation of the cross-reactivities and the effect of sample matrix for the AMEC-based FO-BLI biosensors. (**A**) Evaluation of the assay cross-reactivity towards SARS-CoV, SARS-CoV-2, and MERS; (**B**) Evaluation of the effect of dilution of serum and saliva on RBD measurement.

**Figure 4 sensors-22-03768-f004:**
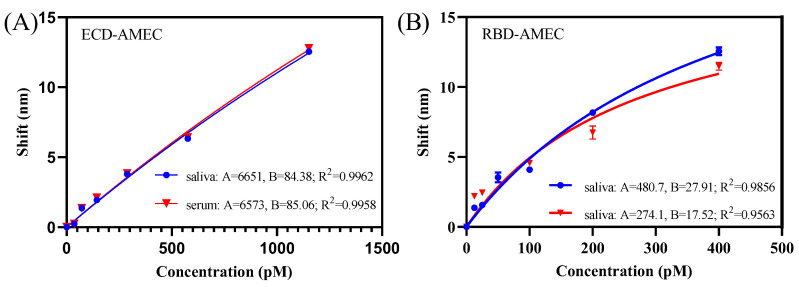
Performance of the AMEC-based FO-BLI biosensor for SARS-CoV-2 RBD and ECD detection in spiked saliva and serum. (**A**) Nonlinear binding curve of ECD detection (36–1152 pM) in high-salt diluted saliva (2-fold) and serum (2-fold); (**B**) Nonlinear binding curve of RBD detection (12.5–400 pM) in high-salt diluted saliva (2-fold) and serum (2-fold).

**Figure 5 sensors-22-03768-f005:**
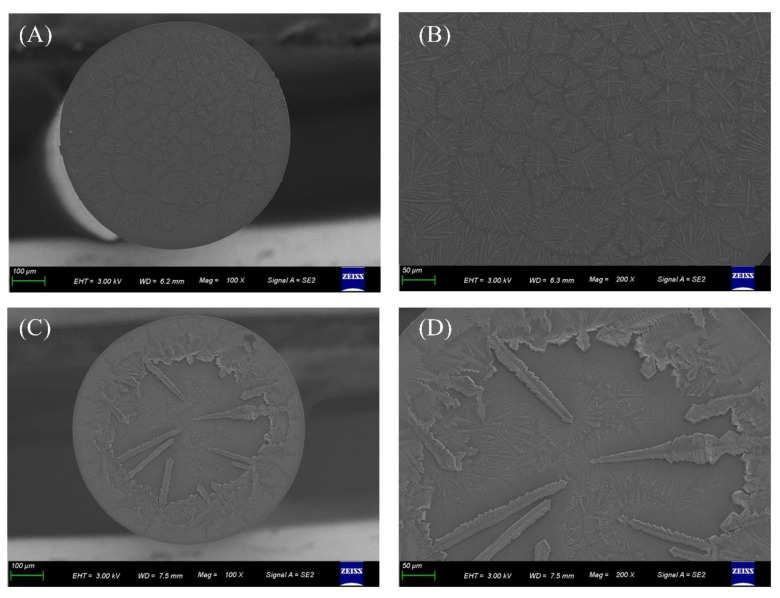
Generation and characterization of the AMEC-induced red precipitates on the optical fiber tips. (**A**–**D**) SEM images of the optical fiber tips when submerged into AMEC solution to react with HRP bonded on the second antibody to attain signals up to 1.4 nm (**A**,**B**) and 16 nm (**C**,**D**).

**Table 1 sensors-22-03768-t001:** Comparison of AMEC and DAB-based FO-BLI biosensors for SARS-CoV-2 detection.

Items	AMEC Based FO-BLI Biosensor	DAB Based FO-BLI Biosensor
Signal enhancer	AMEC	DAB
Detection range for ECD in buffer	36–720 pM	36–720 pM
LoD for pseudovirus	6 × 10^7^	2 × 10^7^
Sample-to-result time	13 min	13 min
Signal-to-noise ratio	14.6	7.4
Stability of enhancer solution	Up to 14 days	30 min
Color of precipitate	Red	Brown
Environmental and human friendliness	▪ Causes serious eye irritation; ▪ No need for a fume hood to operate	▪ May cause cancer and damage to organs; ▪ Toxic to aquatic life with long-lasting effects; ▪ Need for fume hood to operate;
Solubility of precipiate	▪ Insoluble in water but soluble in alcohols and DMF	▪ Insoluble in water and other solvents

**Table 2 sensors-22-03768-t002:** Comparison of performance of the optical biosensors reported in literature and this work for SARS-CoV-2 detection.

Target	Recognition Element	Analytical Sensitivity (LoD)	Time	Substrate	Signal Transducer	Automated	Ref.
RBD	ACE2	12.5 nM	5 s	Carbon Nanotube	Near-Infrared	No	[8]
RBD	Aptamer	37 nM	10 min	D-shaped plastic fiber	SPR	No	[11]
RBD	Antibody	1 pM	3 s	Au/Ag nanostructures	Surface-enhanced Raman spectroscopy	No	[12]
RBD	ACE2	10 pM	30 min	Gold nanoparticles	Colorimetric sandwich bioassay	No	[15]
Omicron	Antibody	More sensitive than PCR	15 min	Plastic fiber	Refractive index variation	No	[16]
RBD	Antibody	33 pM	5 min	Gold	Microcantilever	No	[17]
SARS-CoV-2 sequences	Aptamer	0.22 pM	>25 min	2D gold nanoislands	LSPR	No	[18]
S protein	MIP	58,000 pM	10 min	D-shaped plastic fiber	SPR	No	[19]
RBD, ECD	Antibody	12.5 pM, 36 pM	13 min	Optical fiber	BLI	Yes	This work

ACE2, angiotensin-converting enzyme 2; BLI, biolayer interferometry; ECD, extracellular domain; LoD, the limit of detection; LSPR, localized surface plasmon resonance; MIP, molecularly imprinted polymers; RBD, receptor-binding domain; SPR, surface plasmon resonance.

## Data Availability

Not applicable.

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
