# Peer review of "Rapid Optical Biosensing of SARS-CoV-2 Spike Proteins in Artificial Samples"

_sensors, 2022, doi:10.3390/s22103768_

Round 1

Reviewer 1 Report

The manuscript can be published after a few changes.

I have some comments related to the experimental part

1) On Page 4 line 132, The authors should indicate how the washing step was conducted in more detail.

2) Please insert the structure of AMEC and DAB and the scheme of how they work with the HRP enzyme.

Reviewer 2 Report

This manuscript reported a fiber-optic biolayer interferometry-based biosensor (FO-BLI) to detect the spike proteins related to Covid-19. The performance of low toxic 3-Amino-9-ethylcarbazole substrate was compared with conventional 3,3’-diaminobenzidine (DAB) in terms of sensitivity. Here are some of the questions and suggestions raised after reviewing the manuscript:

  1. Line 64-65, the authors should briefly introduce their previous work such as the functionalization mechanism of the commercial streptavidin SA biosensors and optical detection mechanism.
  2. In Fig. 2A, the authors reported the 6-fold diluted 3-amino-9-75 ethylcarbazole (AMEC) has better assay performance than the pre-defined 200-fold diluted DAB based on signal-to-noise ratio. The authors should mention the definition of “pre-defined 200-fold diluted 3,3’-Diaminobenzidine (DAB)” ? It’s confusion that the authors used different dilution rates for the two types of signal enhancers. Maybe some background should be incorporated here.
  3. As the continue of the previous question, if the authors choose the dilution ratio of DAB randomly, the conclusion could be better supported if the signal shifts of multiple DAB concentrations were presented.
  4. The experiment part lacks repeatability, especially for Figure 2d, it’s hard to conclude the LODs based on single experiment at each condition. Also, why copy number was used when reporting LOD but not molarity?
  5. Figure 4 should include error bars.
  6. Some extra description and modifications are needed in the following:
  • Line 110, “30 mins” should be “30 min”. Also, in this paper, the authors sometimes use minute, please keep it consistent.
  • In section 2.2, “a neutralizing antibody against RDB (MA-RBS)”, here it should be RBD.

Round 2

Reviewer 2 Report

The authors have addressed comments appropriately.